# Rape and the prevalence of hybrids in broadly sympatric species: a case study using albatrosses

Sievert Rohwer, Rebecca B. Harris and Hollie E. Walsh

Department of Biology and Burke Museum of Natural History and Culture, University of Washington, Seattle, WA, USA

## ABSTRACT

Conspecific rape often increases male reproductive success. However, the haste and aggression of forced copulations suggests that males may sometimes rape heterospecific females, thus making rape a likely, but undocumented, source of hybrids between broadly sympatric species. We present evidence that heterospecific rape may be the source of hybrids between Black-footed and Laysan Albatrosses (*Phoebastria nigripes*, and *P. immutabilis*, respectively). Extensive field studies have shown that paired (but not unpaired) males of both of these albatross species use rape as a supplemental reproductive strategy. Between species differences in size, timing of laying, and aggressiveness suggest that Black-footed Albatrosses should be more successful than Laysan Albatrosses in heterospecific rape attempts, and male Black-footed Albatrosses have been observed attempting to force copulations on female Laysan Albatrosses. Nuclear markers showed that the six hybrids we studied were F1s and mitochondrial markers showed that male Black-footed Albatrosses sired all six hybrids. Long-term gene exchange between these species has been from Black-footed Albatrosses into Laysan Albatrosses, suggesting that the siring asymmetry found in our hybrids has long persisted. If hybrids are sired in heterospecific rapes, they presumably would be raised and sexually imprinted on Laysan Albatrosses, and two unmated hybrids in a previous study courted only Laysan Albatrosses.

## INTRODUCTION

Unidirectional hybridization is common in nature. A recent review showed that 50 of 80 cases involving at least five hybrids were predominantly unidirectional (*Wirtz, 1999*). From a long list of alternatives, a shortage of mates for females was the only general explanation supported for unidirectional hybridization. In this paper we seek the beginnings of an answer to the question of why hybrids vary so much in frequency between broadly sympatric species. For example, hybrids between broadly sympatric species of waterfowl and grouse are far more common than they are in other groups of birds (*Grant & Grant, 1992*). Because hybridization usually arises as an epiphenomenon of mating strategies within species (*Price, 2008*), we think hybrids may be disproportionately common in

Corresponding author
Sievert Rohwer, rohwer@uw.edu

groups of birds characterized by forced copulations, as others have suggested (*Kabus, 2002*; *McKee & Pyle, 2002*; *Randler, 2008*). Forced copulations are used as a supplemental reproductive tactic by males in many species of waterfowl (*Brennan et al., 2007*; *McKinney & Evarts, 1998*), but a comparative test by *Randler (2008)* found more support for brood amalgamation than for forced copulations as alternative sources of hybrid waterfowl.

Here we suggest that predicting siring asymmetries offers a promising way to evaluate the importance of heterospecific rape as a source of hybrids between broadly sympatric species. In general, rape supplements male reproductive success when directed toward conspecifics (*Shields & Shields, 1983*; *Thornhill, 1980*; *Thornhill & Palmer, 2001*; *Thornhill & Sauer, 1991*; *Thornhill & Thornhill, 1983*) but the urgent and aggressive nature of rape may result in males sometimes forcing copulations on heterospecific females. While they may be uncommon, hybrids generated by heterospecific rape should be found wherever the parental species breed sympatrically, rather than being confined to zones where the ranges of parapatric species pairs meet and where hybrids are often abundant.

We illustrate our predictions using hybrids between Laysan and Black-footed Albatrosses (*Phoebastria immutabilis* and *P. nigripes*, respectively) because we had genetic samples for both parental species and for hybrids that could be used to test for a siring bias in F1 hybrids and to evaluate long-term gene exchange between the parental species. Paired males of both of these albatrosses are known to force copulations on conspecific females. If hybrids are sired through heterospecific rape, differences between these albatrosses in behavior and the timing of egg laying (detailed below) suggest that Black-footed Albatrosses should sire most F1 hybrids. It is important that only F1 hybrids are used to evaluate siring biases predicted for heterospecific rape because siring asymmetries will be lost if backcross hybrids are generated through random mating with either parental species. Siring bias in F1 hybrids is easily assessed using mitochondrial DNA (mtDNA) to identify the maternal species.

It is important to emphasize that forced and unforced extra pair copulations must be distinguished before the role of heterospecific rape in the generation of hybrids can be assessed. Rape will not be a source of hybrids in species groups where females control extra pair paternity (*Dunn & Cockburn, 1999*; *Spottiswoode & Møller, 2004*; *Stutchbury & Neudorf, 1998*). However, when forced copulations are the result of extreme male aggression, sometimes carried out by groups of males, rape can be a source of hybrids if males mistakenly attack heterospecific females. Of course, male waterfowl have penises that can be used to forcibly inseminate resisting females (*Brennan et al., 2007*; *Brennan, Clarke & Prum, 2010*), but even in species without penises, male rapes may be so aggressive that females must acquiesce to avoid being seriously injured or killed (*Brekke et al., 2013*; *Fisher, 1971*; *McKinney & Evarts, 1998*). Clear evidence of female coercion is required before heterospecific rape appropriately can be considered a possible source of hybrids.

## STUDY SYSTEM

Laysan and Black-footed Albatrosses are closely related sister species (*Nunn et al., 1996*) that breed sympatrically in the Northwestern Hawaiian Islands. Like other albatrosses,

they are long-lived, delay breeding until they are five to seven years old, form life-long pair bonds, lay single eggs, and may breed for 20–50 years (*Fisher, 1969*; *Fisher, 1971*; *Fisher, 1972*; *Fisher, 1975*; *Fisher, 1976*; *Rice & Kenyon, 1962*).

Our samples came from Midway Atoll where over 480,000 pairs of these albatrosses nest and where the beach-nesting Black-footed Albatross comprises about five percent of all pairs (E Flint, pers. comm., 2000). Although interbreeding between Black-footed and Laysan Albatrosses is rare, putative hybrids have been noted for decades (*Fisher, 1948*; *Fisher, 1971*; *McKee & Pyle, 2002*) and up to 20 presumptive hybrids were observed at Midway Atoll between 1997 and 2000 (*McKee & Pyle, 2002*).

Mature Black-footed Albatrosses are primarily dark brown, whereas Laysan Albatrosses are largely white on the body and dark grey to black on the wings and back. Presumed hybrids are intermediate between the parental species in plumage and soft part coloration, ranging in plumage from very pale grey to fairly dark, with pale under wings (*Fisher, 1972*; *McKee & Pyle, 2002*). The lightest presumptive hybrids can resemble the darkest Laysan Albatrosses in plumage color, but the darkest putative hybrids are not as dark as Black-footed Albatrosses. Because both Laysan Albatrosses and hybrids are variable in coloration, identifying or excluding progeny that might result from backcrosses is not possible based on plumage characteristics alone (*McKee & Pyle, 2002*) and requires genetic assessment.

Conspecific rapes are observed in both Laysan and Black-footed Albatrosses (*Fisher, 1971*; *Fisher, 1972*), and Black-footed Albatross males sometimes direct rape attempts at Laysan females, suggesting that hybrids could result from heterospecific rapes. In Laysan Albatrosses conspecific rape is very aggressive, often carried out by groups of males, and sometimes results in serious injury of the female (*Fisher, 1971*). Multiple males regularly join these rape attempts, mounting other males until the pile topples over. Given that albatrosses lack the explosive penis that facilitates forced copulation by male waterfowl (*Brennan et al., 2007*), it seems likely that females may sometimes evert their cloaca to receive sperm just to prevent further harassment and injury by attacking males; however, we should note that *Fisher (1971)* found no evidence of sperm transfer in albatrosses he examined closely following attacks by males. *Fisher (1971)* further reports that he never observed an attempt by the female's mate to defend her from harassing males, as does occur in waterfowl (*McKinney & Evarts, 1998*). Although *Fisher (1972)* reports failing to observe interspecific rape attempts, *McKee & Pyle (2002)* observed male Black-footed Albatrosses attempting to rape female Laysan Albatrosses and believed these events to be the source of hybrids. Neither *Fisher (1972)* nor *McKee & Pyle (2002)* observed mixed pairs attending a nest.

Importantly, differences in the timing of breeding, body size, and aggressiveness all suggest that F1 hybrids are sired when the larger and more aggressive male Black-footed Albatrosses force copulations on female Laysan Albatrosses. Particularly important is that Black-footed Albatrosses arrive at the breeding colonies and lay earlier than do Laysan Albatrosses (*Fisher, 1969*; *Rice & Kenyon, 1962*). Because females take the first incubation shift in these albatrosses (*Fisher, 1971*; *Rice & Kenyon, 1962*), Laysan females are fertile and

vulnerable to insemination through heterospecific rape by Black-footed Albatross males that are mated to females that are already incubating.

We evaluated the F1 status of hybrids using fixed and near-fixed differences in their nuclear genome, and we assessed siring bias using mtDNA from the hybrids. We also used an isolation-migration (IM) model to test the hypothesis of asymmetric gene flow between these species following their divergence approximately 1.03 million years ago (*Nunn et al., 1996*).

## MATERIALS AND METHODS

### Sampling

Blood was sampled from 29 breeding Black-footed Albatrosses, 28 Laysan Albatrosses, and six presumed hybrids (morphologically intermediate between the two species in plumage coloration) at Midway Atoll National Wildlife Refuge (28°13′N, 177°22′W). Genomic DNA was extracted from blood samples either by a standard phenol:chloroform procedure (*Sambrook, Fritsch & Maniatis, 1989*) or using the Wizard SV Genomic DNA Purification System (Promega). All work was conducted in accordance with University of Washington Institutional Animal Care and Use Committee (protocol 2846-13).

### Molecular methods

To assess gene flow between the parental species, we collected DNA sequence data for eight anonymous nuclear loci, one coding nuclear locus (a fragment of a Major Histocompatibility Complex (MHC) gene (*Walsh & Edwards, 2005*), and the mtDNA *cytochrome-b* (*cyt-b*) locus. Anonymous loci were derived from a fosmid library for Black-footed Albatross (Table 1). "FWD" and "REV" designations indicate loci that were taken from opposite ends of a fosmid insert, and therefore are separated by ∼35 kb in the genome. Optimized PCR reactions for anonymous loci contained 0.4 µM primer, 0.2 mM of an equimolar solution of dNTPs, 0.2 U of Taq DNA polymerase (Roche, Indiana, USA), and approximately 20 ng of template DNA in 10 µl reaction volumes. Thermal cycler reaction profiles consisted of initial denaturation at 94 °C for 1 min 30 s, followed by 30 cycles of 94 °C for 30 s, 50–68 °C for 30 s, 72 °C for 45–60 s, and a final extension step of three minutes at 72 °C.

We assigned quality scores to base calls in sequence trace files using Phred (*Ewing & Green, 1998*; *Ewing et al., 1998*) and aligned homologous sequences using Phrap (*Green, 1994*). Polymorphic sites were identified using the program PolyPhred (*Nickerson, Tobe & Taylor, 1997*). Assemblies were visualized in Consed (*Gordon, Abajian & Green, 1998*) and single nucleotide polymorphisms (SNPs) and genotypes at each locus were confirmed by eye. Nuclear haplotypes were resolved using PHASE v.2.1.1 (*Stephens & Donnelly, 2003*; *Stephens, Smith & Donnelly, 2001*). All sequences have been deposited in GenBank (accession numbers KF475302–KF475698).

Putative hybrids were sexed using primers 2550F and 2718R (*Fridolfsson & Ellegren, 1999*); sex was scored by eye, with two bands indicating female and a single-band indicating male. Sexing the hybrids enabled us to assess whether hybrid females (the

**Table 1 Primer and locus information.** Diagnostic nuclear loci (dSNP) that provided at least a 90% probability of distinguishing between the parental species are starred.

| Locus | dSNP | Freq. of dSNP in LA | Forward primer (5′-3′) | Reverse primer (5′-3′) | Length (bp) | %GC | % Identical sites | D BF | D LA |
|---|---|---|---|---|---|---|---|---|---|
| cyt-b | – | – | TTTGCCCTATCTATCCT | GATCCTGTTTCGTGGAGGAAGGT | 609 | 48 | 97.7 | −1.51 | NA |
| MHC* | 1 | 1.0 | CCGGCAGCAGTACGTGCACTTCGNACAGCGA | GATGGGCTGCTGCAGGCTGGTGTGCT | 571 | 63.5 | 99.1 | −0.22 | −1.28 |
| 1FWD* | 2 | 0.90 | GTGCCACCCATGTAAACACCT | TGTGCTTTGGATGAACAGTTG | 429 | 55 | 99.5 | NA | −0.26 |
| 1REV* | 3, 4, 5 | 1.0 | ACTGTGTCACCCCATGCTC | CTGAGTCATTTCCATTCCTGG | 407 | 58.7 | 99.0 | −0.87 | NA |
| 4FWD* | 6 | 1.0 | TGGGCCAGGTTGTTAGGTAG | TATTGGTGGAATGGGCTTGT | 464 | 34.3 | 99.4 | −1.16 | NA |
| 4REV* | 7 | 1.0 | GGCTGGGGGTTTGGAATTA | CTTTCTACAGAGAAATAAACAAAGACC | 443 | 36.9 | 99.5 | −0.24 | NA |
| 6FWD | – | – | AGGGGTCTCTCAAACAGCAA | CTGGCCCTTTAGATAATAGCC | 418 | 35.8 | 99.8 | 1.53 | NA |
| 6REV | – | – | GAAGCGTAGTGAAGTATAACATCGTG | ATGCTGAGGGTGCCATCTTA | 458 | 39.5 | 98.9 | 0.47 | −1.76 |
| 10FWD | – | – | GGCAAAGGCTAAAGGCAAAG | TCAGAATTATTATAGCTTCAGGTGAG | 548 | 43.4 | 99.6 | NA | 0.06 |
| 10REV | – | – | GGTGGTAGAACAGAAAGTCT | TTACCACCTTCCACCACACA | 495 | 36.2 | 99.6 | 0.87 | NA |

**Notes.**
Tajima's D of NA indicates no variation occurring at that locus.
BF, Black-footed Albatross; LA, Laysan Albatross.

**Table 2 Probabilities of F1 and backcross hybrids carrying the observed hybrid genotype.** All six hybrids carried genotype (LA)(A/G)(A/C)(CAG/TGC)(C/T)(A/C); frequencies of the diagnostic SNPs are given in Table 1. The fixed mitochondrial differences render some parental combinations impossible. The shared polymorphism at dSNP 2 makes it possible that the observed hybrid genotype derives from backcrossing, albeit at very low probabilities (<0.05).

| F1 genotype | | |
| --- | --- | --- |
| **F1 combinations** | **mtDNA** | **Probability** |
| LA f × BF m | LA (1.0) | 0.90 |
| LA m × BF f | BF (1.0) | 0.00[*] |
| **Backcross genotype** | | |
| **Backcross combinations** | **mtDNA** | **Probability** |
| F1 f × BF m | LA (1.0) | 0.028 |
| F1 f × LA m | LA (1.0) | 0.034 |
| F1 m × LA f | LA (1.0) | 0.034 |
| F1 m × BF f | BF (1.0) | 0.00[*] |

**Notes.**

[*] Probability is 0 due to the absence of BF mitochondrial haplotype in the observed hybrid genotype.

LA, Laysan Albatross; BF, Black-footed Albatross; f, female; m, male.

heterogametic sex in birds) were inviable, which could be expected under Haldane's rule (*Haldane, 1922*).

## Hybrid identification

We computed two hybrid indices, both varying from 0 (pure Laysan Albatrosses) to 1 (pure Black-footed Albatrosses). The first is most intuitive and includes only loci with fixed or near-fixed sequence differences between Black-footed and Laysan Albatrosses (Table 1). Using these same loci, we also computed the probability that the six phenotypically intermediate specimens were first generation (F1) hybrids or backcrosses (Table 2). For the second hybrid index we used maximum likelihood in the introgress package implemented in R (*Gompert & Buerkle, 2009*), and included all of the nuclear loci.

Siring asymmetries for the hybrids were assessed using a binomial test on mtDNA data.

## Migration estimation

To assess the rate and direction of gene flow between Black-footed and Laysan Albatrosses, we used the IM model implemented in IMa2 (*Hey & Nielsen, 2004*; *Hey, 2010*). We applied the HKY mutation model of nucleotide substitution and nuclear mutation rate scalars were free to vary in the model. The nuclear and mitochondrial genes were assigned an inheritance scalar of 1.0 and 0.25, respectively. To avoid violating the assumptions of no recombination and neutrality of markers, we tested for within-locus recombination using the four-gamete test (*Hudson & Kaplan, 1985*) for each locus and within each species; we tested neutrality of markers using Tajima's D implemented in R package PEGAS (*Paradis, 2010*) (Table 1).

We ran 12 replicate IMa2 analyses, each using different starting seeds and 40–50 concurrent chains, for 10–50 million steps after an initial burn-in phase of 50,000–100,000 generations. To rescale estimates of population size and migration parameters into

**Peer**J

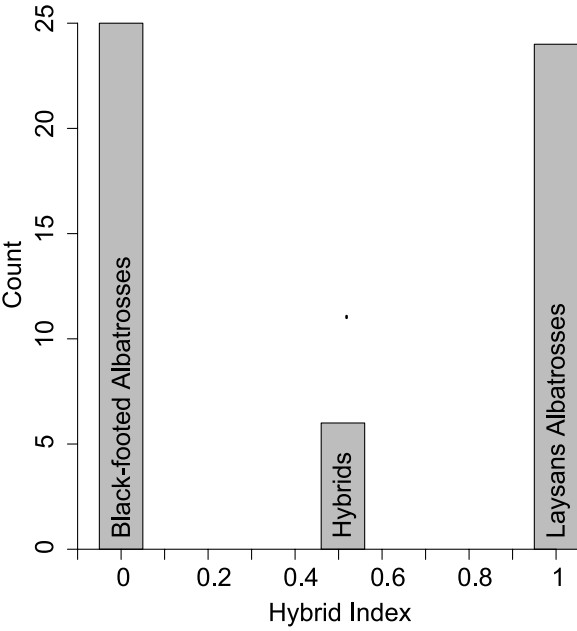

**Figure 1 Hybrid scores based on the five diagnostic SNPS (Table 1).** Pure Black-footed Albatrosses are scored as 0 and pure Laysan Albatrosses are scored as 1. The six putative hybrids all scored as 0.51, rather than 0.50, because Laysan Albatrosses share a rare allele with Black-footed Albatrosses at one of the diagnostic loci.

demographic units, we used the geometric mean of previous mtDNA rate estimates for albatrosses of $3 \times 10^{-5}$ substitutions per locus per year for our fragment of *cyt-b*; (*Nunn & Stanley, 1998*), and a generation time of 25 years (*Cousins & Cooper, 2000*). The results of these independent runs were combined into a single L-mode analysis to evaluate the probabilities for all possible nested models. For explanation of nested models, see the standard IMa2 documentation (*Hey, 2010*). We conducted model selection following (*Carstens, Stoute & Reid, 2009*).

## RESULTS

### Hybrid indices and probability of hybrid genotypes

All six putative hybrids were heterozygous at five diagnostic nuclear SNPs (Table 1). Using just these diagnostic loci the hybrid index for a true F1 hybrid is expected to be 0.51 because Laysan Albatrosses share in low frequency (10%) a single diagnostic SNP (dSNP2 in Tables 1 and 2) that is fixed in Black-footed Albatrosses (Fig. 1).

In Table 2 we use the observed population allele frequencies to calculate the probability of producing the genotype found in all six hybrids, under the assumption that they were either F1 hybrids or first generation backcrosses. The probability of producing the observed hybrid genotype was 0.90 for a parental cross. The probability that the hybrid genotype resulted from a backcross to either of the parental species varies by the sex of the hybrid and the sex and species of the backcross parent (Table 2). Because all hybrids carried Laysan mtDNA haplotypes, the probability of a backcross to a female Black-footed

Albatross is 0. For the three other backcross combinations, the probability of observing the hybrid genotype is either 0.028 or 0.034 (Table 2). These calculations, based on the five diagnostic SNPs, show that the six hybrids are almost certainly F1s and not backcross individuals.

We also evaluated the status of the six hybrids using a maximum likelihood estimator (*Gompert & Buerkle, 2010*), including in this analysis the four nuclear SNPs that were not diagnostic (Table 1). All six hybrids received a score of 0.56, with a 95% confidence interval of 0.22–0.85.

### Siring bias and sex for the hybrids

All six hybrids carried the Laysan Albatross mtDNA haplotype, indicating that F1 hybrids result from male Black-footed Albatrosses inseminating female Laysan Albatrosses ($p = 0.031$). Three of the hybrids were male and three were female, suggesting no inviability of the heterogametic sex (*Haldane, 1922*).

### Gene flow

Tajima's D values showed no significant deviation from neutrality for any of the loci examined (Table 1) and no evidence of recombination within loci was found.

Under the IM model, the rate of gene flow was significantly higher from Black-footed Albatrosses into Laysan Albatrosses ($p = 0.028$). The mean rate of gene flow (2 Nm) was 0.09 gene copies per generation from Black-footed Albatross into Laysan Albatross (95% highest probability density (HPD) 0.024–0.23), whereas this rate was zero in the reverse direction (95% HPD 0–0.10).

Asymmetrical gene flow from Laysan to Black-footed Albatross was constrained to zero in the top four models, which, together, account for 55% of the variation in the weighted AIC (Table 3). A commonly used standard for AIC model ranking is that models within two units of the best model cannot be dismissed. The 5th ranked model does not support unidirectional gene flow (Table 3) and is within two AIC units of the best model. However, this model differs from the best model by one parameter ($k = 3$ vs. 4) and the maximized log-likelihood value of model #5 is similar to that of the best model. This suggests that the larger model #5 is not competitive with the best model and instead is "close" only because it adds one parameter, even though the fit is not improved (Burnham and Anderson, 2002).

## DISCUSSION

Using diagnostic nuclear loci, we show that all six presumed hybrids between Laysan and Black-footed Albatross were F1 hybrids. All six carried Laysan mtDNA haplotypes, indicating that male Black-footed Albatrosses were their sires. This contradicts the hypothesis that a scarcity of mates for females of the rare species results in hybrid pairings (*Wirtz, 1999*) because all six hybrids had Laysan Albatross mothers, instead of mothers of the much less abundant Black-footed Albatross. Finally, we found limited, but significant gene flow from Black-footed Albatrosses into Laysan Albatrosses, suggesting that past F1 hybrids have backcrossed to Laysan Albatrosses. As we discuss below, this further supports our hypothesis that forced copulations are asymmetrical.

**Table 3 AIC ranking of models using IMa2 based on ∼300,000 sampled genealogies.** Model subscripts of population size (q) and migration (m) parameters identify populations used in the analysis; 0, 1, and 2 represent the estimated population sizes for Black-footed Albatrosses, Laysan Albatrosses, and the ancestral population, respectively. In each model brackets denote fixed parameters; other parameters were estimated.

| Model | Log(P) | k | AIC | Delta (AIC) | w | q0 | q1 | q2 | M0>1 | M1>0 |
|---|---|---|---|---|---|---|---|---|---|---|
| Pop. size BF = LA; Mig. from LA to BF = 0 | 2.48 | 3 | 1.04 | 0.00 | 0.16 | 0.2388 | [0.2388] | 0.0085 | [0] | 0.2236 |
| Mig. from LA to BF = 0 | 3.39 | 4 | 1.22 | 0.18 | 0.15 | 0.2244 | 0.07 | 0.00087 | [0] | 2.5594 |
| Anc. pop. size = BF; Mig. from LA to BF = 0 | 2.16 | 3 | 1.68 | 0.64 | 0.12 | 0.3008 | 0.1094 | [0.3008] | [0] | 1.8228 |
| Anc pop. size = LA; Mig. from LA to BF = 0 | 2.16 | 3 | 1.69 | 0.65 | 0.12 | 0.3043 | 0.1101 | [0.1101] | [0] | 1.7566 |
| Mig. from BF to LA = mig. from LA to BF | 2.99 | 4 | 2.03 | 0.99 | 0.10 | 0.2465 | 0.1291 | 0.0026 | 0.1998 | [0.1998] |
| Pop. size LA = BF | 2.48 | 4 | 3.04 | 2.00 | 0.06 | 0.2388 | [0.2388] | 0.0085 | 0 | 0.2236 |
| Mig. from LA to BF = 0; Pop. size LA & BF = anc | 0.30 | 2 | 3.40 | 2.36 | 0.05 | 0.1761 | [0.1761] | [0.1761] | [0] | 1.4231 |
| Anc pop. Size = BF | 2.16 | 4 | 3.68 | 2.64 | 0.04 | 0.3008 | 0.1094 | [0.3008] | 0 | 1.8228 |
| LA pop. size = anc | 2.16 | 4 | 3.69 | 2.65 | 0.04 | 0.3043 | 0.1101 | [0.1101] | 0 | 1.7566 |
| BF pop. size = LA; Mig. from LA to BF = BF to LA | 0.93 | 3 | 4.15 | 3.11 | 0.03 | 0.2388 | [0.2388] | 0.0085 | 0.1034 | [0.1034] |
| Full model | 2.60 | 5 | 4.80 | 3.76 | 0.02 | 0.3394 | 0.1015 | 0.0171 | 0 | 1.6769 |
| BF pop. size = LA & anc | 0.30 | 3 | 5.40 | 4.36 | 0.02 | 0.1761 | [0.1761] | [0.1761] | 0 | 1.4231 |
| BF pop. size = LA; Both mig. = 0 | −0.83 | 2 | 5.66 | 4.62 | 0.02 | 0.2595 | [0.2595] | 0.5181 | [0] | [0] |
| Both mig. = 0 | −0.19 | 3 | 6.38 | 5.34 | 0.01 | 0.2509 | 0.2732 | 0.5181 | [0] | [0] |
| BF pop. size = LA & anc; Mig. From LA to BF = BF to LA | −1.21 | 2 | 6.42 | 5.38 | 0.01 | 0.2907 | [0.2907] | [0.2907] | 0.1121 | [0.1121] |
| LA pop. size = anc; Mig. from LA to BF = BF to LA | −0.69 | 3 | 7.38 | 6.34 | 0.01 | 0.2053 | 0.1214 | [0.1214] | 0.6553 | [0.6553] |
| BF pop. size = anc; Both mig. = 0 | −1.78 | 2 | 7.55 | 6.51 | 0.01 | 0.4489 | 0.3151 | [0.4489] | [0] | [0] |
| BF pop. size = LA; Mig. from BF to LA = 0 | −0.83 | 3 | 7.66 | 6.62 | 0.01 | 0.2595 | [0.2595] | 0.5181 | 0 | [0] |
| BF pop. size = anc; Mig. from LA to BF = BF to LA | −0.85 | 3 | 7.71 | 6.67 | 0.01 | 0.2568 | 0.1392 | [0.2568] | 0.2261 | [0.2261] |
| LA pop. size = anc; Mig. from BF to LA =0 | −0.86 | 3 | 7.73 | 6.69 | 0.01 | 0.4036 | 0.1815 | [0.1815] | 0.1434 | [0] |
| BF pop. size = LA & anc; Mig. from BF to LA = 0 | −2.06 | 2 | 8.13 | 7.09 | 0.00 | 0.2717 | [0.2717] | [0.2717] | 0.4307 | [0] |
| Mig. from BF to LA = 0 | −0.19 | 4 | 8.38 | 7.34 | 0.00 | 0.2509 | 0.2732 | 0.5181 | 0 | [0] |
| BF pop. size = anc; Mig. from BF to LA = 0 | −1.60 | 3 | 9.21 | 8.17 | 0.00 | 0.2659 | 0.0985 | [0.2659] | 1.0792 | [0] |
| LA pop. size = anc; Both mig. = 0 | −3.53 | 2 | 11.05 | 10.01 | 0.00 | 0.2509 | 0.2846 | [0.2846] | [0] | [0] |
| BF pop. size = LA & anc.; Both mig. = 0 | −4.97 | 1 | 11.94 | 10.90 | 0.00 | 0.2644 | [0.2644] | [0.2644] | [0] | [0] |

## Effects of phenology and behavior on insemination biases

Black-footed Albatrosses lay eggs 10 days to two weeks earlier than Laysan Albatrosses (*Fisher, 1969*; *Rice & Kenyon, 1962*), so most female Black-footed Albatross have begun incubating when Laysan females are fertile. This difference in breeding schedules undoubtedly contributes strongly to the asymmetry in inseminations that generate hybrids because only paired males have been reported to engage in rape attempts in these albatrosses (*Fisher, 1971*; *McKee & Pyle, 2002*). Unmated males spend their time at breeding colonies courting females and have not been observed attempting rapes (*Fisher, 1971*). Other factors may also contribute to the observed siring asymmetry. Notably, female Laysan Albatrosses are 5–10% smaller than male Black-footed Albatrosses (*Dunning, 2007*), and male Black-footed Albatrosses are much more aggressive in conspecific rape attempts than are male Laysan Albatrosses (*Fisher, 1972*). Finally, because Black-footed Albatrosses constitute only 5% of the population of these two

species breeding at Midway Atoll, they have far more opportunity to engage in forced heterospecific copulations than do Laysan Albatrosses. These differences suggest that male Black-footed Albatrosses are more likely to sire hybrids through rapes, and all reported heterospecific rape attempts have involved male Black-footed Albatrosses and female Laysan Albatrosses (*McKee & Pyle, 2002*).

The asymmetry in gene exchange revealed by the isolation-migration model suggests a long history of unidirectional gene flow from Black-footed Albatrosses into Laysan Albatrosses. Although modern hybrids appear to have no success in attracting mates (*Fisher, 1972*; *McKee & Pyle, 2002*; *Rice & Kenyon, 1962*), two carefully observed hybrids (unsuccessfully) addressed all courtship attempts at Laysan Albatrosses (*Fisher, 1972*). Hybrids sired by male Black-footed Albatrosses raping female Laysan Albatrosses would be raised by and sexually imprinted on Laysan Albatrosses (*Slagsvold et al., 2002*; *ten Cate & Vos, 1999*) and expected to prefer pairing with Laysans.

## Alternative explanations for asymmetric gene flow

We can think of two alternatives to our hypothesis of heterospecific rape as the cause of the observed asymmetry in gene flow between Black-footed and Laysan Albatrosses. First, is the possibility that F1 backcrosses into the Black-footed Albatross population have not been viable. Definitively addressing this alternative would require breeding experiments, but Fisher's (*1972*) observation that two closely observed hybrids courted only Laysan Albatrosses tends to refute this alternative; although he closely observed just two hybrids, the number of Laysan Albatrosses they attempted to court was large.

Second, if hybrids were intermediate in their breeding schedule relative to the parental species, then hybrids may have had greater opportunity to mate with Laysan Albatrosses, which return later to the breeding colonies than Black-footed Albatrosses. However, this explanation untenably assumes that hybrids form life-long pair-bonds and breed the first year that they return to their breeding islands. Instead, pre-breeding Laysan Albatrosses typically spend one or two years choosing mates (*Fisher, 1972*), making the two-week difference in laying dates unlikely to bias the pattern of backcross matings toward Laysan Albatrosses.

It seems most likely to us that the gene flow revealed by the IM analysis reflects gene exchange that took place as the species were diverging in coloration. This is supported by the fact that courting birds focus their attention on the breasts of their dance partners, where the two species differ most in color (*Fisher, 1972*), and by the failure of field workers to find any hybrids that were paired (*Fisher, 1972*; *McKee & Pyle, 2002*).

## Tests with other groups

The contrast between species in which conspecific extra-pair copulations (EPC) are forced, as opposed to species in which females accept or solicit such copulations, is critical to our thesis that hybrids between broadly sympatric species will be more common in groups where forced copulations are frequent. Although EPC are common in many passerines, they are mostly unforced and apparently controlled by females to increase the genetic

 

quality of offspring (*Dunn & Cockburn, 1998*; *Dunn & Cockburn, 1999*; *Spottiswoode & Møller, 2004*; *Stutchbury & Neudorf, 1998*). Unfortunately, whether EPC are forced or accepted is rarely described in the literature (there are good descriptions of rape in albatrosses, waterfowl, bee-eaters, swallows and the New Zealand Hihi (*Notiomystis cincta*) (*Brekke et al., 2013*; *Emlen & Wrege, 1986*; *Kabus, 2002*; *Martin, 1980*)). Obviously heterospecific rape should not be entertained as a source of hybrids except in groups for which conspecific EPC are clearly forced.

Naturally occurring hybrids are abundant in waterfowl (*Grant & Grant, 1992*; *Randler, 1998*; *Randler, 2008*) and male ducks are known to direct rape attempts at females of other species (*Muñez-Fuentes et al., 2007*; *Randler, 2002*; *Seymour, 1990*). However, we could find no genetic assessments of insemination biases in the generation of hybrids between naturally sympatric waterfowl. An obvious test would be to compare insemination bias when one parental species is characterized by forced copulations and the other is not. For example, hybrids between Northern Shovelers (*Anas clypeata)* and both Mallards (*Anas platyrynchos)* and Northern Pintails (*Anas acuta*) are reported from North America and Eurasia (*McCarthy, 2006*). Because Northern Shoveler males are territorial, and seldom attempt conspecific rapes, the heterospecific rape hypothesis predicts F1 hybrids will have Mallard or Northern Pintail sires (*McKinney & Evarts, 1998*). Siring bias can also be predicted for the abundant hybrids between Common Pochards and Tufted Ducks (*Aythya ferrina* × *Aythya fuligula*, respectively) (*Randler, 2008*). Because conspecific rape is unreported in Common Pochards but frequent in Tufted Ducks (*McKinney & Evarts, 1998*), F1 hybrids should be sired by Tufted Ducks if they were produced through heterospecific rape.

Heterospecific rape probably accounts for the frequent hybrids reported between Barn Swallows (*Hirundo rustica*) and House Martins (*Delichon urbica*) in Europe and between Barn Swallows and Cliff Swallows (*Petrochelidon pyrrhonota*) in North America. Barn Swallows are characterized by many EPCs, but females choose whether or not to accept these EPCs, which are almost never forced (*Møller, 1994*). In contrast, aggressive conspecific rape is frequently observed in both Cliff Swallows and House Martins at communal mud-gathering sites (*Brown & Brown, 1996*; *Møller, 1994*). That male Cliff Swallows and House Martins are characterized by conspecific rape, presumably, renders female Barn Swallows vulnerable to heterospecific rape when they gather mud at sites frequented by males of these two species. Correspondingly, when identified as nestlings, hybrids between Barn Swallows and House Martins were always found in Barn Swallow nests, had Barn Swallow siblings, and had two Barn Swallow parents (*Kabus, 2002*); similarly, nestling hybrids between Barn Swallows and Cliff Swallows or Cave Swallows (*P. fulva*) were found, in all cases but one, in Barn Swallow nests, attended by two Barn Swallow parents (*Martin, 1980*). Given that male Barn Swallows do not force copulations on females, but that males of the three other parental species do force copulations on conspecific females, it seems plausible that most of these were F1s, sired through heterospecific rapes.

## Broader implications

Two comparative studies have addressed the role of EPC in the generation of avian hybrids. In a survey of open nesting birds *Randler (2006)* found EPC to be uncorrelated with the production of hybrids; however, this study failed to distinguish forced and unforced extra pair copulations and failed to consider whether hybrids were rare or common. In another study *Randler (2005)* assessed the roles of forced EPC and brood amalgamation on the production of hybrid waterfowl, and found a significant effect only of brood amalgamation when both factors were included in the model. However, both causal variables were treated as binary characters, which masks their relative importance in species pairs where both factors occur but one generates far more hybrids than the other. Over 800 Common Pochard × Tufted Duck hybrids were reported from Europe (*Randler, 2008*), yet these were treated as equivalent to a single report of a natural hybrid between other species pairs. If most of these 800 hybrids were caused by either factor, then the importance of that factor will be greatly underestimated by failing to account for hybrid frequency.

In some cases evaluating siring asymmetries can generate strong tests of the hypothesis that heterospecific brood parasitism results in ducks forming heterospecific pair bonds (*Randler, 2005*). For example, Redheads (*Aythya americana*) are facultative brood parasites of Canvasbacks (*Aythya valisineria*), whereas Canvasbacks do not parasitize Redhead nests (*Sorenson, Hauber & Derrickson, 2010*). Redhead ducklings raised by Canvasback females in broods of mostly Canvasback ducklings should be sexually imprinted on Canvasbacks and, therefore, be more willing to pair or at least mate with Canvasbacks. Indeed, males of both species cross-fostered into broods predominated by the other species (without hens) preferentially courted heterospecific females (*Sorenson, Hauber & Derrickson, 2010*). An excess of adult males in both species predicts the siring bias: Female Redheads imprinted on Canvasbacks (*Sorenson, Hauber & Derrickson, 2010*), should be able to attract unmated male Canvasbacks as mates. In contrast, male Redheads imprinted on Canvasbacks would be unlikely to attract Canvasback mates because Canvasback females have many unmated males to choose from. Thus Canvasback males should sire F1 hybrids between these species, if hybrids are generated by brood parasitism and sexual imprinting. In contrast, Barrow's (*Bucephala islandica*) and Common (*B. clangula*) Goldeneyes parasitize each other, so males of both species are expected to be sires of hybrids. Although rare, hybrids between both the *Bucephala* and the *Aythya* species pairs are regularly reported (*McCarthy, 2006*), and none of the parentals are characterized by conspecific rape.

Among *Anas* ducks gene sharing through hybridization apparently has strongly affected effective population sizes. For Northern Pintails and Green-winged Teal (*Anas crecca*), census population sizes are too small for certain shared alleles to have persisted for more than 2 and 2.6 million years. But these alleles, which are shared with Mallards, are estimated to have persisted for 6.2 and 7.9 million years, respectively, suggesting a long history of horizontal gene exchange with Mallards, which have a much larger effective population size (*Kraus et al., 2012*). Heterospecific rapes may be responsible for generating F1 hybrids between these ducks and, unlike the situation in albatrosses, F1 hybrid females

in these short-lived ducks may form pair-bonds and breed. Hybrid female ducks should be sexually imprinted on the species that raised them (*ten Cate & Vos, 1999*) and the strong male bias in the breeding sex ratios of north temperate ducks should facilitate pairing and breeding by hybrid females.

## CONCLUSION

Although unidirectional hybridization often predominates in nature, only a shortage of mates for females previously had emerged from a long list of alternative hypotheses as a general explanation for asymmetric hybridization (*Wirtz, 1999*). Here we attempt to make the general point that, if hybrids result from heterospecific rape, differences in behavior and life history of the parental species can be used to predict the direction of crosses. Predicting the mother and father species of F1 hybrids from different parental combinations has the potential to considerably refine our understanding of the importance of heterospecific forced copulation (and brood parasitism), in the generation of hybrids (*McKee & Pyle, 2002*; *Møller, 1994*; *Randler, 2005*). Although heterospecific rape is unlikely to be adaptive, it has the potential to explain differences in the prevalence of F1 hybrids between broadly sympatric species pairs according to whether or not they are characterized by conspecific forced copulations.

Several authors have suggested that heterospecific rape may be an important source of avian hybrids (*McKee & Pyle, 2002*; *Møller, 1994*; *Randler, 2005*), but Randler's (*2005*) comparative study of waterfowl found only weak support for this hypothesis. We believe that testing for siring asymmetries will provide a stronger assessment of this hypothesis in waterfowl, a group for which wild hybrids have been reported between many pairs of broadly sympatric species (*Grant & Grant, 1992*). Forced copulations have been reported for various insects (*Arnqvist, 1989*; *Thornhill, 1980*; *Thornhill & Sauer, 1991*), fish (*Valero, Garcia & Magurran, 2008*), lizards (*Cooper, 1985*; *Olsson, 1995*; *Rodda, 1992*) and mammals (*Harris et al., 2010*), but whether or not forced copulations generate hybrids in these groups has not yet been addressed.

### Postscript

Coincident with our revision of this manuscript Hope Ronco and Pete Leary (US Fish & Wildlife Service) informed us of a hybrid albatross (Fig. 2) at Midway Atoll that is paired with a Laysan Albatross and that has successfully raised chicks several times since 2006. Its sex is unknown because they have not observed it mating. As far as we know this is the first record of a Black-footed x Laysan Albatross hybrid successfully breeding. Of course, it is only a single bird, but that it is mated to a Laysan Albatross is consistent with the hypothesis that its sire would have been a Black-footed Albatross and that it would have been raised and imprinted on Laysan parents. Its apparent success at raising backcross chicks with a Laysan is also consistent with the asymmetry in gene flow suggested by our IM analyses. Blood samples to confirm that it is an F1 hybrid, and blood samples from its chicks would add valuable additional information to this remarkable observation.

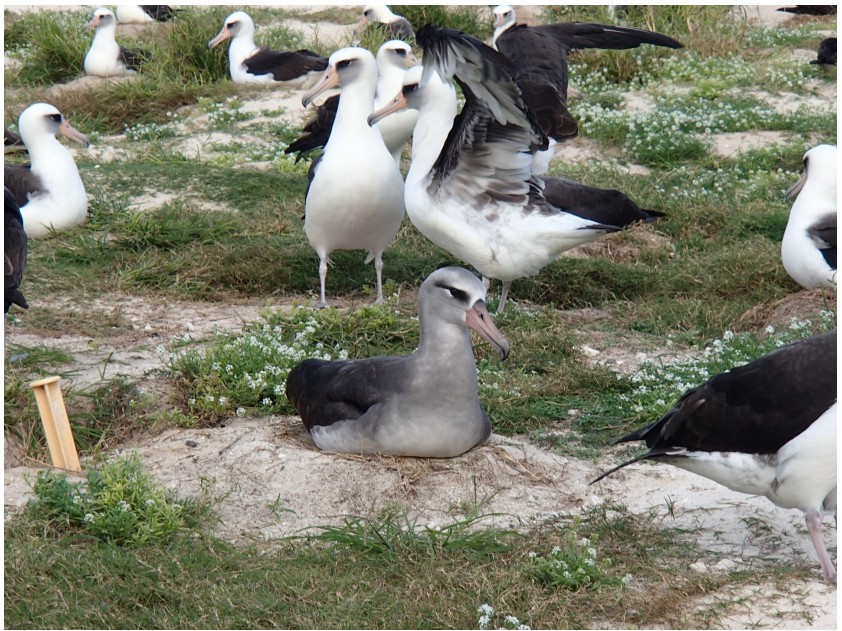

**Figure 2 A recently documented hybrid that is mated to a Laysan Albatross and has raised chicks.** H. Ronco of the USFWS provided the photo.

## ACKNOWLEDGEMENTS

Nancy Hoffman and Peter Pyle collected and provided hybrid blood samples for this study, and S Edwards provided blood samples for black-footed and Laysan Albatrosses. E Flint and N Hoffman engaged in helpful discussions. A Møller provided information on swallows and helped with references. W Swanson provided workspace, and C Elphick, P Brennan, J Walsh, J Felsenstein, S Edwards, and members of the Rohwer, Leaché and Klicka lab groups commented on the manuscript. Thanks to all for their help.

### Funding

An NSF dissertation improvement grant, an NSERC PGS B postgraduate scholarship, and the Leaders Five Endowed Fellowship of the UW Burke Museum supported HEW for this project. The funders had no role in study design, data collection and analysis, decision to publish, or preparation of the manuscript.

### Grant Disclosures

The following grant information was disclosed by the authors:
An NSF dissertation improvement grant.
An NSERC PGS B postgraduate scholarship.
Leaders Five Endowed Fellowship of the UW Burke Museum.

### Competing Interests

The authors declare there are no competing interests.

## Author Contributions

- Sievert Rohwer, Rebecca B. Harris and Hollie E. Walsh conceived and designed the experiments, performed the experiments, analyzed the data, contributed reagents/materials/analysis tools, wrote the paper, prepared figures and/or tables, reviewed drafts of the paper.

## Animal Ethics

The following information was supplied relating to ethical approvals (i.e., approving body and any reference numbers):

University of Washington Animal Care and Use Committee, Approval Number 2846-13.

## Field Study Permissions

The following information was supplied relating to field study approvals (i.e., approving body and any reference numbers):

U.S. Fish and Wildlife Service U.S. License and Permit Nos: MB755833-0 and 12521-00013.

## DNA Deposition

The following information was supplied regarding the deposition of DNA sequences:

GenBank KF475302–KF475698.

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
