# Peer review of "Rape and the prevalence of hybrids in broadly sympatric species: a case study using albatrosses"

_PeerJ, doi:10.7717/peerj.409_

## Round 0.1 · original submission · Major Revisions

This is an interesting paper and both reviewers saw merits in aspects of the study. Both also raised important concerns that must be addressed before the paper is suitable for publication. Although I would like to see all comments thoroughly addressed, those of particular importance are (a) the lack of evidence that the hybrids arose from forced copulations (especially in light of concerns about the role of females raised by reviewer 1), (b) the small sample sizes, which greatly limit the statistical inferences that can be made, (c) more complete information on the molecular methods, (d) greater clarity in the presentation of results in the tables (and perhaps elimination of certain tables, see reviewer 2; although given that space is not limiting on-line, I see not problem with providing the extra information as long as it is clearly presented), and (e) more tempered conclusions given the limited data set. I anticipate that these changes will require substantial modifications to the content and tone of the current manuscript.

·

Basic reporting

The article meets all the standards

Experimental design

I am not a population geneticist and therefore cannot comment at all on the suitability of their genetic methods employed to determine the direction of the hybridization, and the likelihood that these are all F-1 rather than backcross individuals.

Validity of the findings

The sample size of hybrids analyzed is small (n=6). Could the authors collect samples from museum specimens that may have hybrids in order to increase their sample size?

Additional comments

There is no description presented on how these forced copulations take place. In ducks, many forced copulations are successful because they are carried out by groups of males that can overpower the female’s mate, and also because male ducks have penises that they can use to forcefully inseminate females even when she is resisting. Because Albatross only lay one egg, I would imagine females should resist forced copulations quite strongly, and her mate should defend the female when a forced copulation attempt is taking place. Also, females in species where males do not have a penis, must evert their cloaca to receive male sperm, so why would female albatross evert their cloaca for force copulating males? More details as to whether this is acquiescence to harassment, and the role of the female’s mate are needed in the introduction.

Line 199: delete been

222: There is something missing here. I do not understand the argument…

244-254: It is true that there is heterospecific rape in ducks, however, ducks often form heterospecific pair bonds as a result of brood parasitism, and therefore it is impossible to know whether any particular waterfowl hybrid was sired because a female pair-bonded with a male of the wrong species, or because she was raped. In fact this was one of the main findings of Randler’s paper… that hybrids were most often correlated with IBP, rather than FEPCs. Though I agree with the commentary later on problems with his analysis, the main observation is correct: ducks form pair bonds with other species quite often.

280-281. For the reason stated above, I think that unidirectional pair bonds would also have the same results. This could easily happen if females prefer supernormal stimuli and preferentially bond with males that offer a more intense display (I believe Randler 2002 may have discussed this as well).

305-306. I disagree. Randler 2005 tested this hypothesis in waterfowl quite extensively (though perhaps not using the best approach).

·

Basic reporting

• The background literature review could be better developed. The role of forced copulations in these species is not accurately described according to the literature cited. Fisher 1972 suggests that while forced copulation could lead to hybridization, no forced copulation had been observed between the two species over the duration of their field study. They also questioned the success of intraspecific forced copulations. Do the authors have their own observations to supplement this? Or are there other papers that document inter-specific forced copulations? Similarly, the theory behind unidirectional hybridization needs further development (e.g. see meta-analysis in Randall 2005 on the importance of forced copulations relative to other explanations). There could be alternative explanations for the occurrence of F1 hybrids. As well, the authors’ treatment of forced copulation ignores the potential role of female choice (e.g. solicitation male competition, controlling paternity, sperm competition, etc.). A more thorough review of previous genetic work identifying hybrids and backcrossing events in this system would also be helpful. Line 70 mentions the use of molecular methods to assess backcrossed individuals. A reference to this work or a summary would help clarify the context of the current research.
• My biggest concern with the paper is that the data collected do not actually test the predictions made by the authors. While the authors did document F1 hybrids between black-footed and Laysan albatrosses and showed that all F1 crosses had mitochondrial DNA of the Laysan albatross, the authors have no observational field data to show the F1 individuals in question were the product of forced copulations. Nor do they have any current or direct observational data of forced copulation between heterospecifics on their study site. One could speculate on the mechanisms, including potentially forced interspecific copulations, responsible for the observation of unidirectional hybridization, but the data do not speak directly to any of those mechanisms.
• I would suggest limiting the scope of this paper to genetic documentation of F1 hybrids with results consistent with unidirectional hybridization (but see sample size and methodological limitations below). Accordingly, content about forced copulations should be reduced and the title changed to more accurately reflect the findings.
• Throughout, replace the word “rape” with “forced copulation”. Rape is rarely used in current animal behavior literature.
• I don’t think table one is necessary. The hybrid index provides strong enough support for the 6 individuals being F1 hybrids. Table one is difficult to follow and does not add much to the paper. Table two was also difficult to follow and lacks critical information on model names/descriptions. Table three should provide primer information for MHC and cytochrome b (and references for cytochrome b primers).

Experimental design

The paper presents interesting documentation of F1 hybrids in this system. Genetic identification of hybrids in this systems appears to be limited prior to this study, in which case the results presented by the researchers are important for understanding hybridization events between black-footed and Laysan albatrosses. Concerns/comments on the experimental design are as follows:
• I wonder what conclusions can be made about the backcrossing and asymmetrical geneflow given the observational evidence in the literature that F1 hybrids appear to be generally unsuccessful. Backcrossing appears to be a rare event, and I worry that the signal being picked up on by the IM models could be unreliable (e.g. an artifact).
• IM may not preform well with the small sample sizes used in this study and the output from this approach is notoriously difficult to interpret (IM documentation- https://bio.cst.temple.edu/~hey/program_files/IM/Introduction_to_IM_and_IMa_3_5_2007.pdf). The confidence intervals for the estimates of gene flow from one species to another are overlapping, presenting challenges with the IM interpretation.
• Although informative, the sample size for putative hybrids is low. Choosing 6 putative hybrids based on plumage characteristics (which are reported to be highly variable in these species) limits the conclusions that can be made. A thorough sampling of the breeding grounds would be necessary to rule out reciprocal hybridization. Nonetheless, the findings for these six hybrids are novel and interesting on their own (possibly as a short note).
• Based on the description of the population sampled, it appears that the unidirectional hybridization could still be based on a numbers game. The rare black-footed albatross males may be more likely to engage in heterospecific mating due to lack of available black-footed females. How variable are the relative numbers of black-footed and Laysan albatrosses within and across breeding grounds? Might the patterns of unidirectional hybridization vary by site?
• I suggest further interpretation and description of Haldane’s rule. While equal numbers of male and female F1 hybrids proves that females are not inviable, it does not discount the possibility that female hybrids are less fit compared to males.
• A summary of PCR conditions should be included (at minimum, magnesium concentrations and annealing temperatures for each primer).
• Description of molecular markers is difficult to follow. Which markers were diagnostic? How was this conclusion reached? Have the markers been previously validated? Has the relevant info on these markers been published elsewhere?

Validity of the findings

• In general, the results do not strongly support the conclusions reached. Conclusions of forced copulations are speculative (and should be identified as such), as no data were collected to evaluate this. While the data support unidirectional hybridization, a sampled size of 6 F1 hybrids with intermediate plumage from one location limits the conclusions that can be made system-wide.

---

## Round 0.2 · Major Revisions

Thank you for your revision. First I should apologize for the long time it has taken to make a decision. Given your disagreement with several of the more substantive reviewer comments, I felt it important to evaluate each point carefully and finding time to do that took longer than I anticipated. I find merit in some of your arguments, but also find more merit in the reviews than you perhaps give them credit for. At this time, I do not believe the paper is suitable for publication, but I do think it could be quite easily and invite another round of revision should you be willing to address my comments, as described below. My comments follow the order of your response letter.

1. In your response you state that your fundamental goal is only to seek “the beginnings of an answer to the question of why hybrids vary so much in frequency between broadly sympatric”. Nonetheless, language in the discussion implies (to me, at least) that you are doing more than this – and in my view go beyond the evidence (e.g., the statement “heterospecific rapes are likely responsible for generating F1 hybrids” in ducks, among several examples). I suspect that this is the basis for the reviewer concerns. Recasting these claims more clearly as untested propositions could easily resolve the differences. Equally, stating the “fundamental point of the paper” more clearly in the introduction (i.e., along the lines that it is stated in the response letter) might help.

2. I agree with you that the exact test does support a difference. But a small sample size still warrants great caution when it comes to interpretation. Type I errors remain a concern, especially when just one more individual could change the inference. Indeed, this issue is one that has received substantial attention in the recent literature on the difficulty that scientists have replicating statistically significant results.

3. In response to the reviewers’ concerns that your data do not test your predictions, you argue that, although you cannot prove that heterospecific rape is the cause of unidirectional hybridization, to meet the goals of this paper it is sufficient that your data are consistent with that hypothesis. This is a fair point, as long as it is acknowledged that the test is not an especially strong one. The problem I see – and the one that I suspect resulted in the reviewers’ concerns, is that the paper goes beyond suggesting that the data are consistent with the hypothesis. E.g., the abstract states “We show heterospecific rape to be the probable source of hybrids …”, the start of the conclusions also suggests that hybrid albatross “result from heterospecific rape”, etc. Yet, there is no evidence presented to address alternative hypotheses, such as the possibility that a small number of female albatrosses choose heterospecific pairings (e.g., perhaps because an egg got laid in the wrong nest and the resulting chick was raised by parents of the other species, which it then imprinted on – I’m not saying this is a likely scenario, just that alternatives of this type are not considered). For the paper to be accepted, the manuscript text needs to be consistent in tone/caution with the arguments made in the response letter.

4. I have gone back and forth over the issue of whether “rape” is an appropriate term to use. Like the reviewers, my inclination is to err away from it because of the difficulty of inferring motivations and because of the term’s human connotations. After reviewing various definitions, however, I’m not sure I can justify my instinctual stance. Moreover, “forced copulation” (which is what I too would have suggested as an alternative) suffers from the same problems concerning motivations. So, I leave this to your discretion as I do not feel that current conventions in the literature are sufficient reason for me to object.

5. I do not know enough about IM analysis to judge whether your sample sizes are adequate. I would, however, say that the fact that others have published based on similar sample sizes is not a good enough justification – plenty of work gets published in which the statistics are questionable. Are there theoretical grounds for considering your sample adequate? A response on this issue that directly addresses the paper cited by the reviewer would be helpful.

6. Perhaps my biggest concern about the results relates to the model comparisons. Table 3 only presents a small subset of the models compared and without seeing the entire model set it is difficult to make inferences about which variables are more important than others. Moreover, the 6 models shown all have ΔAICs <2 and low Akaike weights, suggesting that they are functionally equivalent in their ability to explain the data and that none of these models really stands out over the others. Given that the models shown appear to include both models with unidirectional and bidirectional gene flow (if I am reading the first column correctly), the statement that “Our results show that models with unidirectional gene flow are consistently ranked the highest” is not very well supported (i.e., within this group there is no basis for concluding that the 5th model is any better than the 1st). This conclusion is consistent with the inferences one would make from the overlapping confidence intervals. Moreover, if the next X models are only marginally worse than these 6 (say, all have ΔAIC <4-5, then the story could be even murkier). Although ΔAIC < 2 is a common cut-off, it is a debatable one and it is helpful to know more about the full set of models. To feel confident about the conclusion here I would want to see, at least, all models with delta AIC < 10. Model-averaging across all models considered (not just the subset shown) might also provide a clearer signal, and is an appropriate and standard extension of the information theoretic multi-model approach.

7. This comment from the reviewers was not addressed in the response and I do not see text changes that address it. Please clarify if I am missing something:
• A summary of PCR conditions should be included (at minimum, magnesium concentrations and annealing temperatures for each primer).

8. The legend for Tables 1 has been used for Table 2 as well.
Finally, I have made a number of additional comments (many minor editorial points) directly on the manuscript, using track changes to facilitate your ability to examine them. I will send this file under separate cover as PeerJ has no mechanism for me to attach a document to this response letter.

Despite my lengthy comments I continue to find this paper to be an interesting contribution and hope that you will feel comfortable making the suggested changes. Should you decide to submit a further revision, please address each of the comments in this letter, and those on the manuscript in your response.

---

## Round 0.3 · accepted · Accept

Than you for your detailed revision, and for bearing with my lengthy comments and slow turnaround. At this point I am happy to accept your paper, pending only the minor editorial corrections listed below. Please also double check your manuscript for species name capitalization and the references one last time.

Abstract, 5th line: capitalize “Black-footed”
p. 4, 4th line in last paragraph of introduction: “extra pair” should be two words (for consistency with other uses within the paper).
p.7, 3rd line in “Hybrid Identification” section: capitalize “Black-footed”. Also, lower case “identification” for consistency with other subheads at this level.
p. 8, 1st line in “Migration estimation” section: capitalize “Black-footed”. (Maybe search on this word to look for other instances.)
p. 8, last but one line in Methods: you have “(cite IMA2 manual)”. I presume you meant to actually do so.
p. 8, First subhead in Results: Do not capitalize “indices”.
p. 10, two lines from the top, end of penultimate Results paragraph: Define HPD – the acronym, not the statistical concept. If this is just a credible interval, then maybe use that term instead, since it is more widely known.
p. 10, end of 2nd paragraph: Capitalize the “Albatross” immediately before the Slaggsvold et al. reference.
p. 10, 2nd line of section of Alternative explanations section: Capitalize “Black-footed”
p. 12, last paragraph of of Alternative explanations section: “attention ON the ..”
p. 12, 1st line of “tests with other groups” section: I don’t see a need to capitalize “extra-pair copulations”; it’s not normally treated as a proper noun.
p. 13, end of first full paragraph: “Tufted Ducks” should be capitalized.
p. 14, end of top paragraph: “Swallows do no FORCE copulations”
p. 14, last line: second instance of “Canvasbacks” in sentence should also be capitalized.
p. 15, top line: first instance of “Redhead” (before the Sorenson reference) should be capitalized.
p. 15, 9 lines down in first paragraph: another uncapitalized “Canvasback”
p. 15, end of first paragraph: “parentals ARE characterized …”
p. 15, last paragraph: two instances of “Mallard” that need capitalizing.
p. 17: First Ewing reference: “Research” in journal name should be capitalized.
p. 17: Second Fisher reference: Are there journal and page numbers?
p. 18: First Gompert reference: “Ecology” in journal name should be capitalized.
p. 20, Stephens reference: “Bayesian” should be capitalized, since it derives from someone’s name.
Table 2 legend: Should say “are given IN Table 1”
Table 3: either add an opening square brackets around the q1 value, or remove the closing bracket, whichever is appropriate.